# Effects of Aquatic versus Land High-Intensity Interval Training on Acute Cardiometabolic and Perceptive Responses in Healthy Young Women

**DOI:** 10.3390/ijerph192416761

**Published:** 2022-12-14

**Authors:** Manny M. Y. Kwok, Eric T. C. Poon, Shamay S. M. Ng, Matthew C. Y. Lai, Billy C. L. So

**Affiliations:** 1Gait and Motion Analysis Laboratory, Department of Rehabilitation Sciences, The Hong Kong Polytechnic University, Hong Kong, China; 2Department of Health and Physical Education, The Education University of Hong Kong, Hong Kong, China

**Keywords:** interval physical exercise, water immersion, aerobic fitness, psychological health, comma

## Abstract

The effects of aquatic high-intensity interval training (AHIIT) on cardiometabolic and perceptive responses when compared to similar land-based exercise (LHIIT) remain unknown. Here, we aimed to (1) establish a matched intensity between mediums and (2) compare the acute cardiometabolic and perceptive responses to the two interventions in healthy young women. Twenty healthy young women performed a stationary running exercise at a matched exercise intensity. The incremental stages, in terms of percentage of heart rate (HR), maximal oxygen uptake (%VO_2_max), percentage of oxygen uptake reserve (%VO_2_R), percentage of heart rate reserve (%HRR), and rate of perceived exertion (RPE), were examined and acute cardiometabolic and perceptive responses were evaluated. The results showed that HR was significantly reduced (AHIIT: W 150 ± 19, R 140 ± 18, LHIIT: W 167 ± 16, R 158 ± 16 *p* < 0.01) and oxygen pulse (AHIIT: W 12 ± 2, R 10 ± 2, LHIIT: W 11 ± 2, R 9 ± 2 *p* < 0.05) was significantly increased with AHIIT compared to LHIIT. No significant group differences were observed for the perceptive responses. The comparable results demonstrated by the aquatic and land incremental tests allow precise AHIIT and LHIIT prescriptions. AHIIT had distinct differences in HR and oxygen pulse, despite having no distinct difference from LHIIT for some cardiometabolic and affective responses.

## 1. Introduction

Physical exercises are beneficial to health. Exercise has been shown to reduce the risk of a number of cardiometabolic diseases, including cardiovascular diseases, cancer, and diabetes [1]. Given this evidence, the WHO has recommended that all adults should undertake 150–300 min of moderate-intensity, or 75–150 min of vigorous-intensity physical activity, or some equivalent combination of both per week [2]. However, around one-in-three women and one-in-four adults failed to achieve physical activities up to or exceeding the WHO recommended level [3]. One of the most common barriers to exercise faced by women was lack of time, probably due to career commitments or family responsibilities [4]. Hence, women have a higher cardiometabolic risk (CMR), which should raise public health concerns [5]. An effective exercise regimen that has a reduced time commitment has the potential to encourage more participation in exercise, thereby decreasing the CMR associated with physical inactivity. 

High-intensity interval training (HIIT) has been defined as repeated short bouts of high levels of exercise (≥80% maximal heart rate) interspersed with periods of rest or low to moderate exercise levels, which lasts for less than 30 min per training [6]. It has been promoted as a more time-efficient and effective option than continuous training for improving cardiometabolic health, weight management, and both insulin and blood glucose regulation [7].

Based on the increasing popularity of aquatic exercises, predominantly among women, performing HIIT in an aquatic environment (AHIIT) is an alternative to land-based HIIT (LHIIT) [8]. AHIIT has the potential to be even more beneficial than LHIIT, because of the additional physiological benefits associated with the physical properties of water [9]. For example, water buoyancy reduces the discomfort and stress placed on the joints. The hydrodynamic nature of water can act as a form of resistance to movements, thereby optimizing the development of muscle strength [10]. Evidence also supports the benefits of AHIIT for improving cardiometabolic health and aerobic capacity (VO_2_) in non-athletic populations and women when compared to no exercise training [11,12]. AHIIT therefore represent an option to minimize the physical inactivity associated with a higher CMR in women.

In addition to the potential physiological benefits brought by AHIIT, it has been suggested that exercise can also have benefits for the perceptive responses that are congruent to exercise compliance [13]. Perceived pleasure during exercise can lead to behavioral changes in physical activity, and thus perceptive response is an important factor for decision-making in exercise prescription [14]. Enjoyment can be described as a positive effective state that reflects feelings such as pleasure, liking, and fun, and that is associated with physical activity participation and adherence [15]. A study assessing HIIT interspersed with rest periods showed the development of positive perceptions, resulting in a higher exercise compliance [16]. For instance, it was suggested that an increase in perceived affect was associated with an additional 38 min of physical activity per week [17]. Identifying perceptive responses plays an essential role in improving exercise adherence. 

In order to the optimize training outcomes from both AHIIT and LHIIT, identifying the intensity of exercise needed for any individual is crucial [18]. The application of land-based intensity could underestimate metabolic demand in water [19]. Although the physiological parameters of heart rate (HR) and oxygen capacity (VO_2_) response are often used to monitor exercise intensity, the assessment of these parameters during water immersion is challenging [20]. Hence, an individualized exercise intensity corresponding to incremental tests should be closely monitored and adhered to in land and water [21]. 

The physiological responses during aquatic- and land-based exercise may differ. The research that has been conducted to date has provided inconsistent findings with respect to the cardiometabolic responses to land- and water-based exercises, when those exercises were performed at maximal exertion. For example, a number of studies used the same cycling or treadmill walking exercise performed in water and land, and showed similar levels of maximal aerobic capacity (VO_2_ max) between the two types of exercise [22,23]. However, other studies reported a lower aerobic power and ventilatory capacity when comparing aquatic running to land running programs [24,25]. Hence, to date, little evidence has been found comparing AHIIT with LHIIT for cardiometabolic response differences. Similarly, the perceptive responses of AHIIT remain to be determined when compared to LHIIT. Knowledge regarding the effects of both AHIIT and LHIIT—both in terms of how they are similar and how they differ—is important, in order to understand when one might be recommended over the other.

Given these considerations, the current study had two specific aims. First, we sought to identify a matched level of exercise intensity needed for AHIIT and LHIIT in a sample of healthy women. Second, we sought to make a direct comparison between AHIIT and LHIIT, with respect to their acute effects on measures of cardiometabolic and perceptive responses. Given the evidence cited above, regarding the potential greater benefits of AHIIT relative to LHIIT, we hypothesized that AHIIT would result in higher responses, in measures of both cardiometabolic response and positive perceptive responses.

## 2. Material and Methods

### 2.1. Participants

Twenty young healthy and active women were recruited through local poster advertisement from the university community. The inclusion criteria were women who were non-pregnant, clinically healthy, and between 20 and 35 years of age. The exclusion criteria included subjects with chronic medical and health conditions, fear of water, and skin diseases. All participants were informed of the study risks by an investigator and signed an informed consent form prior to data collection. They were then administered an international physical activity questionnaire, to assess their physical activity levels for descriptive purposes [26].

Using G*power software and based on the effect size 0.64 obtained, assuming a power of 0.8 at an alpha level of 0.05 (G*Power version 3.0.10), the sample size computed was 16 or more subjects per group [27]. Considering an estimated 20% attrition rate, the anticipated sample size of 20 subjects per condition was adequate for detecting differences between groups. The study was approved by the office of research ethical committee, the Hong Kong Polytechnic University (HSEARS20210522001). This study conformed to the Declaration of Helsinki for studies involving humans. 

### 2.2. Study Design and Procedures

A randomized crossover design was used in this study. The 20 participants were randomly assigned, based on an online random number generator, to complete AHIIT and LHIIT trials. Participants were asked not to complete any aerobic training for 48 h prior to any testing sessions, to limit the influence of prior exercise on the study outcomes. Each participant attended two sessions at the pool and two sessions on land at the laboratory of our institution. Incremental testing was completed, first either at the pool or on land, followed by the AHIIT and LHIIT. For AHIIT, immersion was at chest depth (xiphoid-sternal depth or up to five centimeters deeper) in a swimming pool (temperature 29 °C). For LHIIT, the room temperature of ambient air was maintained at 23 °C.

### 2.3. Aquatic and Land Incremental Tests

An incremental tests in water and on land were performed prior to the exercise interventions, to confirm an individualized cadence required at a matched level of exercise intensity (stationary running at 90% with 1 min active recovery at 70% HRmax in between) in each condition. We first measured the participants’ anthropometric data, including (1) body weight in kg and (2) body height in cm with an electronic scale (BC-730b, Tanita, Japan) and a stadiometer, respectively. The incremental test was carried out using stationary running. Instructions for stationary running directed participants to flex the hip and knee to as close to 90° as comfort and control allowed and to then push to straighten the hip and knee. Prior to testing, all exercises were demonstrated, then practiced once. Participants were monitored continuously and recorded at a frequency of 1 Hz using a HR sensor (Polar OH1, Kempele, Finland). The HR sensor used has been shown to provide valid and reliable HR data [28]. During the incremental test, gas exchange data were obtained using a portable metabolic device PNOE. The PNOE device was operated in a breath-by-breath mode, which continuously measures volume and simultaneously determines expired gas concentrations. It was calibrated prior to each session according to the manufacturer’s specifications. PNOE was validated in previous research, as compared to a validated stationary metabolic cart (COSMED QUARK-CPET) [29]. The incremental protocol increased the exercise load from 85 beats per minute (bpm) and increased the cadence by 15 bpm every 2 min for each progression [24]. A metronome (IMT 300, Tokyo, Japan) was used to monitor the speed of movements during the trial. The HR, VO_2_, and rate of perceived exertion (RPE) per minute were recorded. VO_2_max was considered to be attained when the following standardized criteria were met: (1) a respiratory exchange ratio of greater than or equal to 1.10; (2) failure of heart rate to increase with increases in workload; (3) post-exercise blood lactate ≥8.0 mmol·L^−1^ [30]; (4) clear signs of exhaustion (facial flushing, unsteady gait); and (5) refusal to carry on despite strong verbal encouragement. Maximal oxygen capacity (VO_2_max) was determined using both aquatic and land incremental tests, with stationary running to test to volitional exhaustion. HR, percentage of VO_2_ max (%VO_2_max), percentage of HR max (% HRmax), percentage of VO_2_ reserve (%VO_2_R), and percentage of HR reserve (%HRR) were recorded for the two environments. Blood lactate was measured via capillary blood sampling from the fingertips with a portable analyzer (Lactate Plus, Nova Biomedical, Waltham, MA, USA). Data collected from the incremental test were used to determine the intensity required for the exercise interventions for each participant.

### 2.4. AHIIT and LHIIT Exercise Interventions

Participants performed a standardized 3 min warm-up at 50% HRmax before exercises. The instructor then asked the participants to run in place at the cadence determined from the prior incremental tests. AHIIT or LHIIT training was performed under the desired cadence with an audible metronome. The AHIIT and LHIIT protocol consisted of 10 bouts of 1 min stationary running at 90% HRmax, separated by 1 min active recovery at 70% HRmax. The total exercise trial time for each condition was 20 min. The two tests were separated by at least 48 h and a maximum of 72 h. All the sessions were held at the same time of day, to avoid variations related to circadian rhythms. No external stimuli, such as music or verbal encouragement, were provided during either intervention. The outcomes were compared with the matched intensity performed in the exercise interventions. 

### 2.5. Primary Cardiometabolic Outcomes

For the AHIIT and LHIIT trials, VO_2_, oxygen pulse, respiratory exchange ratio (RER), minute ventilation (VE), and HR were measured using a PNOE device. 

### 2.6. Secondary Cardiometabolic Outcomes

In addition, measures of participants’ energy expenditures (EE), cumulative EEs, and metabolic equivalent (MET) were used in both estimation procedures, with measurement via indirect calorimetry using PNOE. Blood lactate concentrations were recorded immediately before and after the AHIIT or LHIIT in all trials. Capillary blood samples (approximately 25 µL) were acquired from the fingertips using a portable analyzer (Lactate Plus, Nova Biomedical, Waltham, MA, USA).

### 2.7. Secondary Perceptive Responses

#### 2.7.1. Enjoyment

Exercise enjoyment was assessed using an 18 item Physical Activity Enjoyment Scale (PACES) scale immediately after each intervention. With the PACES, respondents indicated their level of enjoyment for 18 different items, using a 7 point bipolar Likert scale, ranging from 1 (“I enjoy it”) to 7 (“I hate it”). The possible range was 0 to 126, and higher scores indicated a greater level of enjoyment during the exercise intervention. Evidence supports the reliability and validity of the PACES for assessing enjoyment of exercise when engaged in physical activity [31].

#### 2.7.2. Self-Efficacy

Self-efficacy is conceptualized as beliefs relative to one’s capabilities to successfully execute a necessary course of action [32]. Participants’ self-efficacy was assessed via a 5 item questionnaire, designed to determine participants’ confidence to repeat either AHIIT or LHIIT. This self-efficacy scale has been demonstrated to have good internal consistency (α’s = 0.9) [33]. Responses were scored at a percentage of 0% (not at all) to 100% (extremely confident) with 10% increments, and then averaged for the five items. Participants were asked to complete the scale immediately after each intervention. 

#### 2.7.3. Muscle Soreness

Muscle soreness was measured with the use of a 7 point Likert scale of muscle soreness for lower limbs, which combines verbal and numeric cues. Indeed, it is suggested that Likert-based scales are easier to use, require a shorter time to explain to patients, and do not require anchoring procedures, which may be influenced by the experience of the subjects. The construct validity of the Likert scale as a measure of lower limb muscle soreness is well supported [34].

### 2.8. Statistical Analysis

Descriptive statistics were first computed for the demographic data and study variables for descriptive purposes. Next, we conducted a series of Shapiro–Wilk tests, to evaluate the normalcy of the distributions of data. Continuous data measures were then summarized with means and SDs. To assess the main effect of medium, main effect of stages, and interaction effect of medium and stages between aquatic and land incremental tests, a repeated-measures ANOVA was used for comparisons of HR, %HRmax, VO_2_, %VO_2_max, %VO_2_R, %HRR, and RPE at different incremental stages. A paired t-test was used to compare the within group differences for the cardiometabolic and perceptive variables with AHIIT and LHIIT. All continuous data were used for statistical analysis, with a significance level of *p* < 0.05. All analyses were conducted using IBM SPSS Version 25.0 (IBM Corporation, Armonk, NY, USA) software.

## 3. Results

### 3.1. Description of the Study Sample and Study Variables

The descriptive statistics for the study sample are presented in Table 1, and the number of participants who reached each stage of the incremental test in the aquatic and land environments is presented in Table 2. The mean age of the participants was 21.95 ± 2.35 yrs. Their average height was 160.95 ± 5.76 cm, and average weight was 53.95 ± 8.08 kg. The physical activity levels deduced from IPAQ were 85% at level 2 and 15% at level 3. All the participants were able to complete the four stages of the incremental test in both environments.

### 3.2. Effects of Increased Exercise Intensity on HR, %HRmax, %VO_2_max, %HRR, VO_2_, %VO_2_R, and RPE in the Aquatic and Land Incremental Tests

With an increase in the incremental stages of cadence, there was an increase in the physiological responses for HR, %HRmax, %HRR, VO_2_, %VO_2_max, %VO_2_R, and RPE (Figure 1A–G). As illustrated in Figure 1A–F, there was a significant main effect of the incremental stages (*p* < 0.001) on the physiological variables (HR, %HRmax, %HRR, VO_2_, %VO_2_max, %VO_2_R), as well as the subjective RPE (*p* < 0.001). There was a significant main effect of medium on HR, %HRR, and RPE (*p* < 0.05) (Figure 1A,C,G). There was no significant interaction effect of water immersion found for HR, %HRmax, VO_2_, %VO_2_max, % HRR, and %VO_2_R and subjective RPE (*p* = 0.053, *p* = 0.169, *p* = 0.308, *p* = 0.828, *p* = 0.893, *p* = 0.873, *p* = 0.868, respectively). The descriptive intensity between the aquatic and land incremental tests is shown in Table 3.

### 3.3. Effects of AHIIT and LHIIT on Primary Cardiometabolic Outcomes

The AHIIT group showed a significant decrease in HR and HRmax, in both work and recovery intervals (*p* < 0.01). The AHIIT HR and HRmax were significantly lower when compared to the LHIIT. The AHIIT oxygen pulse was significantly higher than that from LHIIT (*p* = 0.038). There were no significant differences in the other cardiorespiratory parameters between AHIIT and LHIIT (Table 4).

### 3.4. Effects of AHIIT and LHIIT on Secondary Cardiometabolic Outcomes

The EE, MET, and cumulative EE did not show any significant differences (*p* > 0.05). There was also no significant difference in the blood lactate concentration change between AHIIT (6.08 ± 2.86 mmol/L) and LHIIT (5.84 ± 2.42 mmol/L) (Table 4).

### 3.5. Effects of AHIIT and LHIIT on Perceptive Responses

There were no significant differences found for the RPE of work (*p* = 0.6) and recovery intervals (*p* = 0.948) for AHIIT and LHIIT. Both AHIIT and LHIIT responded similarly for enjoyment (*p* = 0.875), self-efficacy score (*p* = 0.072), and muscle soreness index (*p* = 0.873) (Table 5). 

## 4. Discussion

To the best of the authors’ knowledge, this is the first study to determine the effect of AHIIT and LHIIT on the acute cardiometabolic and perceptive responses among young women. One major objective of this study was to examine whether an acute bout of AHIIT would stimulate greater cardiometabolic and perceptive responses than LHIIT. Our principal finding was that AHIIT was associated with a lower HR, HRmax, and higher oxygen pulse, with a similar perceptive response relative to LHIIT in healthy women.

This study examined acute cardiometabolic and perceptive responses to AHIIT and LHIIT with the same cadence intensity, determined in aquatic and land incremental tests. This indicates that the exercise trials were of similar relative intensity within subjects, and differences in outcomes between exercise bouts were a function of the medium difference. From our findings for the incremental tests performed in both environments, as the exercise cadence increased to different incremental stages, the variables (HR, %HRmax, VO_2_, %VO_2_max, %HRR, %VO_2_R, and RPE) increased. This is in agreement with previous studies, which highlighted that as the speed increased in both aquatic and land environments, the cadence, HR, and VO_2_ also increased [35]. Despite there being no significant interactions between the medium and incremental stages for the variables (HR, %HRmax, %HRR, VO_2_, %VO_2_max, %VO_2_R, and RPE), the interaction-effect of the medium and incremental stages for HR approached significance (*p* = 0.053). We suggest that the use of aquatic and land incremental tests allowed for a direct and specific examination of the effects of water immersion. Based on the present findings, instructors and coaches may use the intensity revealed between aquatic and land incremental tests to efficiently and precisely prescribe HIIT training sessions. Thus, when it is not possible to directly measure the aforementioned variables with incremental tests, as in the practical conditions in gyms and clubs, matched intensity may be used to individualize the aerobic load prescription.

This study confirmed a reduction in HRmax, work, and recovery HR during aquatic AHIIT compared to LHIIT, which supports our hypothesis. It also provides more evidence of the accuracy of exercise prescription when intensity was monitored solely using HR. Our findings were similar to previous aquatic- and land-based studies, where the heart rate decreased with water immersion [36]. It is known that with water immersion, hydrostatic pressure causes the blood to become displaced from the peripheries to the center, resulting a significant increase in the venous return and volume of the heart. This mechanism promotes a reduction in HR because of stimulation of the carotid and aortic receptors, which is likely to be directly proportional to the immersion depth [9]. Therefore extra caution should be taken when using the HR values obtained on land to regulate exercise in water, since exercise intensity was one of the variables for aquatic exercise prescription [37]. 

A significant increase in oxygen pulse was demonstrated in AHIIT compared to LHIIT in the work phase. Although there was an insignificant change in the recovery phase, the *p* value was marginal and approached significance (*p* = 0.078). The oxygen pulse provides a reflection of the oxygen taken up by the pulmonary blood during the period of a heartbeat, the combined product of stroke volume, and the difference between the arterial and mixed venous blood oxygen contents [38]. The increase in the oxygen pulse can be attributed to the increase in arteriovenous oxygen difference. This could be explained by the more effective ventilation at peak exercise in the aquatic environment, as a result of decreases in physiological dead space, which, in turn, caused an increased in cardiorespiratory efficiency during the progressively increasing exercise work rate under immersion [39]. Another possible reason for the higher oxygen pulse found in the AHIIT in our study could be the increased breathing frequency that occurs while submerged in water at the chest level [40]. A comparatively greater ventilatory drive may have been required to overcome the effects of hydrostatic pressure on the thoracic cavity, causing an increased residual volume and decreased tidal volume and vital capacity. As a result, a higher oxygen pulse was revealed in AHIIT than LHIIT in our study, which could potentially further challenge the cardiorespiratory fitness of individuals.

The rest of the cardiorespiratory outcomes (VO_2_, VO_2_max, %VO_2_max, RER, VE) differed insignificantly between the AHIIT and LHIIT trials. The VO_2_ max was not different and this was also observed in Masumoto et al. (2007) and Greene et al. (2011), who compared maximal tests on a treadmill in both environments [41,42]. Similarly, Silver et al. (2007) demonstrated no significant differences between the VO_2_ max in incremental treadmill tests in both environments [40]. Alberton et al. (2009) suggested stationary running did not produce a significant change in VO_2_ max and proposed that the VO_2_ max depends on the muscle mass involved [22]. The water environment might have reduced the VO_2_ max by reducing the vital capacity, total lung capacity, and pulmonary elasticity, which caused VO_2_ to be consumed by the respiratory muscles and reduced the VO_2_ availability to other muscles, and hence reduced the overall contribution to VO_2_ max. This agrees with previous review findings with stationary water running and stationary running on land and with water cycling and a bicycle ergometer [24]. Therefore, a determining factor for the VO_2_ max pattern is the mode of exercise performed, rather than the inherent physical properties of water.

In our study, the total EE for a 20 min AHIIT or LHIIT intervention was between 610 and 667 kcals (7–9 kcals/min; 7–9 METs). Our findings indicated that the EE of the HIIT programs was similar to those reported in previous studies involving aquatic interval trainings [43]. The comparatively high EE achieved could be explained by the increased speed and intensity of optimized water resistance causing a higher EE, while maintaining the same range of movements [44]. Our results were in line with previous aquatic- and land-based studies, where HIIT elicited a comparatively greater EE when compared with constant-intensity or continuous exercise regimes [25]. 

The mean values of blood lactate in AHIIT were similar to immersed running compared to the values observed in LHIIT. The blood lactate level was the net lactate difference between the lactate production and elimination, which was positively associated with post exercise fatigue [45]. A possible reason for our result of no significant difference between the two mediums was that we adopted interval physical exercise, with active recovery consisting of 60 s. The active recovery period could have compensated for the energy consumed and facilitated the removal of metabolites. Therefore, this may have helped to decrease the post-exercise lactate concentration and offset the influence of the environment on the production of lactate. This was also supported by the study conducted by Chien et al. (2020); when AHIIT and LHIIT were compared, there was no difference in the lactate level between the two environments immediately post exercise [46].

As for the perceptual variables, no significant group difference for RPE, enjoyment, self-efficacy, and muscle soreness was found. HIIT combines time efficiency, diversity, and fun. The physiological benefits brought by HIIT have been widely published and are aligned, while the perceptive responses in HIIT have not reached a consensus among authors [47]. Our result is in agreement with Ma et al. (2017), who suggested that both aquatic and land interval training elicited similar changes in RPE in a group of women [48]. It was also suggested that AHIIT was perceived as being more affective and enjoyable than LHIIT in men with obesity [49]. This variation in results may be related to variability in water depth, subject characteristics, genders, exercise intensity, intervals, and depend on the modalities performed in AHIIT.

This study has several strengths. The novelty of an incremental test performed prior to the HIIT intervention allowed the correct, optimal, and matched intensity of HR, %HRmax, %HRR, VO_2_, %VO_2_max, %VO_2_R, RPE, and monitoring in AHIIT and LHIIT, for comparison. In most previous land and water comparison studies, the expressed intensity was based on incremental maximum tests performed on land [50]. In this study, the intensity was based on the values obtained from an aquatic incremental test, which has been shown to be the most accurate and appropriate methodological option for exercise prescription in the aquatic environment, because of the aforementioned characteristics and properties of water [51]. The matched intensity provides a precise guideline for determining an individual’s baseline level of fitness for AHIIT prescriptive purposes and will serve as a method of outcome assessment for AHIIT programs. 

Despite these strengths, the major limitations of the present study include that only young female participants were recruited, and hence caution should be taken when generalizing to older women, as well as men. Some other limitations in this study were that the subjects belonged to healthy populations, and it might not be possible to generalize to clinical populations. Although running movement is a basic and natural exercise for humans, it may not be representative of all the exercises. In addition, the design of this study was a cross-sectional study, which examined the acute effect of HIIT interventions only, it was not representative of long-term effects. Nevertheless, our results provide practical guidelines for applying matched-intensity aquatic and land incremental tests, followed by corresponding HIIT interventions. From a practical point of view, AHIIT can be an adjunct or alternative to land-based HIIT, for improving the cardiometabolic response and enhancing perceptual responses in women. This may be ideal for women who are unable to exercise on land or those who exclusively train on land and want to cross-train in water for assessment or rehabilitative purposes. By evaluating cardiometabolic outcomes, therapists or exercise professionals are able to better promote the unique benefits and physiologic advantages of AHIIT and LHIIT. This may lead to the engagement of women to participate in AHIIT or LHIIT, as a more widely practiced form of physical activity. A randomized control study could be adopted to study the long-term effects of the interventions on physiological or perceptual outcomes among the different environments. Therefore, future studies examining the efficacy of AHIIT using clinical populations, athletes, and those unable to perform LHIIT are also warranted.

## 5. Conclusions

In summary, our findings suggest that the comparable results demonstrated by the aquatic and land incremental tests allowed participants to perform exercise in the same domain. Moreover, AHIIT had distinct differences from LHIIT for heart rate and oxygen pulse, despite there being no distinct differences in some cardiometabolic and perceptual variables, at least immediately after an acute bout of exercise. This suggests that AHIIT can offer cardiometabolic benefits and perceptual responses comparable to LHIIT. Future research with randomized controlled trials of a longer duration are suggested to compare the effects of AHIIT and LHIIT on cardiometabolic and perceptual outcomes.

## Figures and Tables

**Figure 1 ijerph-19-16761-f001:**
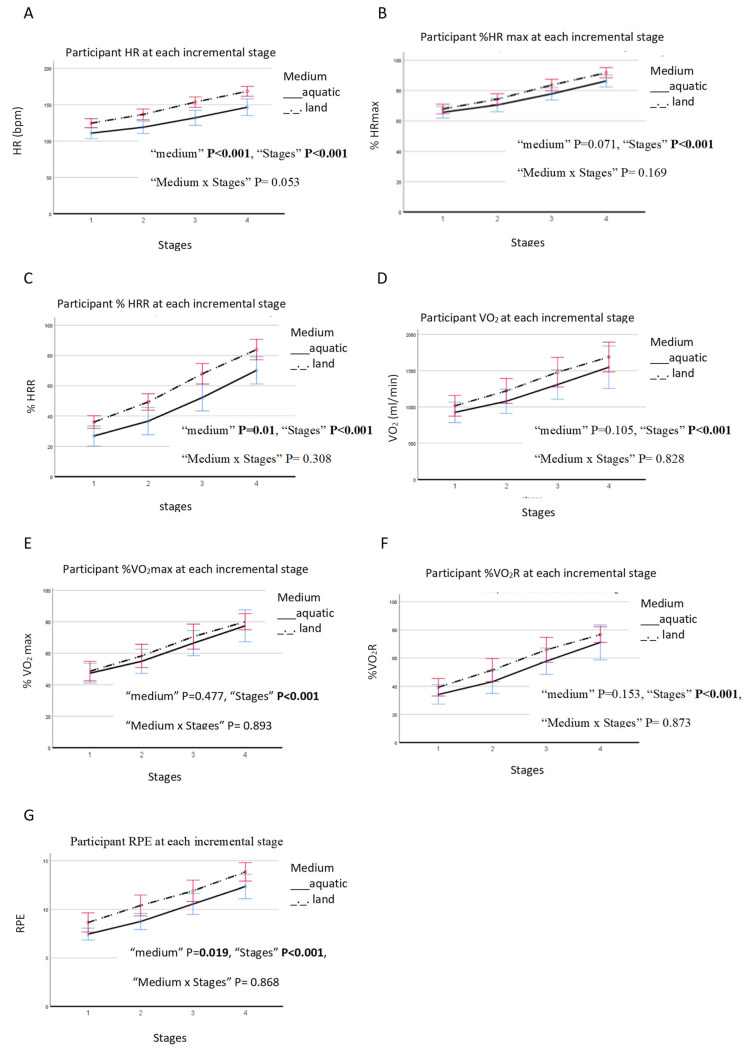
Aquatic VS land incremental tests: (**A**) HR, (**B**) %maxHR, (**C**) %HRR, (**D**) VO_2_, (**E**) %VO_2_max, (**F**) %VO_2_R, and (**G**) RPE at the different incremental test stages.

**Table 1 ijerph-19-16761-t001:** Descriptive characteristics of participants: (mean ± SD).

	Participants
N Sex Age (in years) Height (cm) Weight (kg) International Physical activity levels (%) Level 1 (inactive) Level 2 (minimally active) Level 3 (active)	20 F 21.95 ± 2.35 160.95 ± 5.76 53.95 ± 8.08 0% 85% 15%

**Table 2 ijerph-19-16761-t002:** Number of participants who reached each stage of the aquatic and land incremental tests.

Stage Number	Cadence (bpm)	Aquatic Incremental: N (%)	Land Incremental: N (%)
1	85	20 (100%)	20 (100%)
2	100	20 (100%)	20 (100%)
3	115	20 (100%)	20 (100%)
4	130	20 (100%)	20 (100%)
5	145	19 (95%)	16 (80%)
6	160	11 (55%)	12 (60%)
7	175	9 (45%)	8 (40%)
8	190	8 (40%)	7 (35%)
9	205	6 (30%)	3 (15%)
10	220	4 (20%)	0 (0%)

**Table 3 ijerph-19-16761-t003:** Descriptive statistics (mean ± SD) for HR, %HRmax, %HRR, VO_2_, %VO_2_max, %VO_2_R, and RPE at each stage of the aquatic and land incremental tests.

Stages	Medium	HR	%HRmax	%HRR	VO_2_	%VO_2_max	%VO_2_R	RPE
1	Aquatic Land	110.9 ± 16.2 136.7 ± 15.8	65.7 ± 8.0 67.8 ± 6.7	26.8 ± 14.1 36.1 ± 8.9	926.0 ± 303.8 1016.7 ± 305.1	47.4 ± 13.4 49.6 ± 13.2	34.1 ± 14.7 39.3 ± 13.3	7.5 ± 1.3 8.7 ± 2.1
2	Aquatic Land	119.0 ± 18.2 153.6 ± 15.3	70.5 ± 9.4 74.3 ± 7.5	36.6 ± 19.3 49.2 ± 11.7	1078.3 ± 357.3 1221.4 ± 366.7	54.9 ± 16.4 58.3 ± 15.8	43.2 ± 17.6 51.5 ± 17.5	8.8 ± 1.8 10.4 ± 2.3
3	Aquatic Land	132.0 ± 21.9 168.5 ± 14.5	77.8 ± 8.6 83.6 ± 8.1	52.2 ± 19.0 67.9 ± 14.5	1309.2 ± 431.7 1479.8 ± 436.3	65.5 ± 17.1 70.6 ± 17.0	57.8 ± 20.0 65.9 ± 18.9	10.6 ± 2.3 11.9 ± 2.3
4	Aquatic Land	146.6 ± 24.5 168.5 ± 14.5	86.3 ± 8.4 91.7 ± 7.2	70.2 ± 19.2 83.9 ± 14.5	1548.6 ± 625.3 1689.7 ± 438.7	77.4 ± 21.6 80.0 ± 11.0	71.1 ± 26.5 76.7 ± 11.9	12.4 ± 2.7 13.9 ± 2.0

Note: HR = heart rate; VO_2_ max = maximal oxygen uptake; HRmax = maximal heart rate; RPE = rate of perceived exertion; HRR = heart rate reserve; VO_2_R = VO_2_ reserve.

**Table 4 ijerph-19-16761-t004:** Cardiometabolic outcomes in the AHIIT and LHIIT interventions (mean ±SD).

Measure	AHIIT	LHIIT	*p*-Value
HRmax (bpm)	162 ± 19.1	179.1 ± 14.3	<0.01 *
HR (bpm)	W149.62 ± 18.88 R 139.26 ± 17.90	W166.75 ± 16.41 R158.07 ± 15.78	<0.01 * <0.01 *
%HRmax (%)	W92.71 ± 4.25 R 86.40 ± 6.60	W93.26 ± 3.98 R 88.75 ± 3.47	0.693 0.177
VO_2_max (mL·kg^−1^·min^−1^)	35.78 ± 6.58	36.14 ± 7.24	0.819
VO_2_ (mL·min^−1^)	W1758.57 ± 348.76 R1383.19 ± 280.37	W1829.18 ± 287.67 R1462.29 ± 318.55	0.293 0.273
%VO_2_max (%)	W91.4 ± 5.98 R72.18 ± 10.15	W95.38 ± 11.29 R76.66 ± 11.02	0.091 0.173
Oxygen pulse (mL/beat)	W11.81 ± 2.05 R9.92 ± 1.55	W11.05 ± 1.78 R9.27 ± 1.78	0.038 * 0.078
RER	W0.93 ± 0.09 R1.05 ± 0.09	W0.94 ± 0.06 R1.01 ± 0.08	0.460 0.178
VE (L/min)	W61.31 ± 15.31 R52.40 ± 13.63	W62.93 ± 11.22 R51.35 ± 8.81	0.621 0.704
EE (kcal/min)	W8.64 ± 1.70 R6.99 ± 1.44	W9.02 ± 1.44 R7.33 ± 1.57	0.257 0.340
MET	W9.22 ± 1.50 R7.28 ± 1.48	W9.79 ± 1.52 R7.77 ± 1.35	0.122 0.189
Cumulative EE (kcal) Lactate changes (mmol/L)	W609.81 ± 235.53 R663.98 ± 255.72 6.08 ± 2.86	W618.70 ± 246.65 R667.07 ± 265.39 5.84 ± 2.42	0.897 0.967 0.572

Notes: W = work period, R = active recovery period, VO_2_ = oxygen uptake, VO_2_max = maximal oxygen uptake, RER = respiratory exchange ratio, VE = minute ventilation, EE = energy expenditure, MET = metabolic equivalent. * (*p* < 0.05).

**Table 5 ijerph-19-16761-t005:** Perception outcomes for AHIIT and LHIIT (mean ± SD).

Measure	AHIIT	LHIIT	*p* Value
RPE (6–20)	W13.18 ± 2.0 R11.66 ± 2.18	W12.86 ± 1.84 R11.71 ± 1.89	0.60 0.948
Enjoyment (Score of 126)	68.55 ± 7.53	68.55 ± 9.24	1.00
Self-efficacy (Score of 100)	37.8 ± 27.48	45.75 ± 20.91	0.072
Muscle soreness index (Score 0–6)	5.3 ± 2.07	5.4 ± 1.7	0.873

Note: RPE, rate of perceived exertion.

## Data Availability

The datasets used and/or analyzed during the current study are available from the corresponding author on reasonable request.

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
