# Peer review of "Effects of Aquatic versus Land High-Intensity Interval Training on Acute Cardiometabolic and Perceptive Responses in Healthy Young Women"

_ijerph, 2022, doi:10.3390/ijerph192416761_

Round 1

Reviewer 1 Report

The paper is quite interesting, well documented and correctly compiled, and the references are currently update. However, I think it is suitable for publication after clarifying the following aspects:

1. Please see the journal's guidelines for article preparation (IJERPH Microsoft Word template file). These specify that in the abstract you must remove words: Background, Methods, Results, Conclusions.

2.  Key words: "Interval exercise" - exercise is a much too general term and I recommend specifying the fact that it is interval physical exercise. In this context, I think „physical activity” should be removed from the keywords so as not to repeat itself!

3. At 2.1. Participants, you specified that it is a group of women between the ages of 20 and 35. As you specified in table 1, the mean age of the subjects is 21.95±2.35. As such, I think you should specify in the title that it is about young women.

4.    In 2.2. Study design and procedures, you specified that each subject participated in two sessions in the pool and two sessions on land, and Table 3 shows data collected after 4 training sessions (4 in the aquatic environment and 4 on land). Is this a typo (drafting error) or did I get it wrong? Please clarify this point.

5.    The data presented in table 2 is not clear enough and I would like you to clarify it:

-       In the paper you specify that it is HIIT (High Intensity Interval Training), and maximum intensity implies a HR higher than 180 bpm. The subjects reach this value only on the 8th repetition (out of the 10 programmed), the first 5 being performed at low and medium intensity, and the 6th and 7th repetitions at submaximal intensity. In this context, I don’t understand the term High Intensity......

-  Also, in chapter 2.2 you mentioned that "The 20 participants were assigned to complete AHIIT and LHIIT trials, with order randomly assigned", and from the tabular data it appears that not all of them completed these training programs. What happened to the subjects? Did they stop? In this case, such an approach does not capture essential aspects, such as level of tolerance to effort/subject, physical condition/subject, pleasure/subject, etc. Please clarify this aspect!

-  From the data presented in table 3, it appears that the cadence increased with each repetition, which means that the work was done progressively. But this is another type of training called "progressive effort interval training" and it is not the same as HIIT. Please clarify this and modify everywhere where HIIT is concerned.

-  At stage number 7, only 9 subjects reached the respective HR in the aquatic test and only 8 in the land test. Ditto for series 8,9 and 10. What happened to the other subjects (respectively 55%)? They enter in the statistics and for this reason, I consider it distorted, for all the parameters targeted. If you only address to the maximal intensities, as the title of the study suggests, logically only subjects who managed to reach this level of effort should have been counted on the two working methodologies: aquatic/land. As a result, I suggest you clarify the physical training procedure used and adapt to it both the title, the purpose of the study and then, accordingly, the statistical interpretations.

6.   References:

-       I did not find the first reference - Chestnov O. World Health Organization Global Action Plan for the Prevention and Control of Noncommunicable Diseases. Geneva, Switzerland. 2013. Is found only „Global Action Plan for the Prevention and Control of Noncommunicable Diseases. Geneva, Switzerland” to the address: https://www.who.int/publications/i/item/9789241506236   or https://apps.who.int/gb/ebwha/pdf_files/WHA66/A66_R10-en.pdf?ua=1

Is that correct? Please clarify this aspect!

-  According to the recommendations requested by the editors, references should be indicated by a numeral or numerals in square brackets—e.g., [1] or [2,3], or [4–6]. Please correct this aspect in the full-text!

-  I recommend that the references be reviewed because they do not fully meet the requirements. No DOI was included for all references where it exists.

Overall, I felt that the study needed clarification regarding the training methodology applied.

Best regards!

Author Response

Dear reviewer,

Thank you for your valuable comments and time. We tried our best to address your comments and hope can make the paper clearer for the readers. Please kindly see attached. 

Thank you very much.

Best Wishes!

Reviewer 2 Report

The paper is important and well done. There are minor syntactical changes:

Line 60 "Responses that are"

Lines 65-67: revise to "A study assessing HIIT interspersed with rest periods has shown development of positive perception of that exercise technique, which resulted in improved exercise compliance. (If there are multiple studies, these will also need referencing, but the article only cit4s a single article.)

Line 82: water and land have shown

Line 84: "when comparing"

Line 118: the randomization process needs description.

Lines 186-187: I'm confused. If the Likert scale goes from 1 ("best") to 7 (worst"), how can a higher score represent greater enjoyment levels?

Line 231: "created on"

Line 298: "AHIIT compared to LHIIT"

Line 308" "pulse resulted found in AHIIT"

Author Response

Dear reviewer,

Thank you so much for your comments and time. We are grateful for your advice. We hope things are clearer after addressing your comments. Please kindly see attached response. 

Thank you.

Best Wishes!

Round 2

Reviewer 1 Report

I consider the article suitable for publication in this form.

Best regards!